# LncRNA H19 Promotes Lung Adenocarcinoma Progression via Binding to Mutant p53 R175H

**DOI:** 10.3390/cancers14184486

**Published:** 2022-09-16

**Authors:** Yaodong Zhou, Qing Xia

**Affiliations:** 1Department of Thoracic Surgery, Fudan University Shanghai Cancer Center, Shanghai 200032, China; 2Department of Neonate, Children’s Hospital of Fudan University, Shanghai 201102, China

**Keywords:** lung adenocarcinoma, H19, p53, gene–gene interaction, VMRC-LCD

## Abstract

**Simple Summary:**

This research explored the association and interaction between lncRNA H19 and mutant p53 (R175H) in lung adenocarcinoma development and progression. H19 over-expression may induce the elevated expression of mtp53 and interact with mtp53, which prolongs the p53 half-life and promotes transcriptional activity, leading to the progression of lung adenocarcinoma. The simultaneous inhibition of H19 and mtp53 may provide a novel strategy.

**Abstract:**

Background: Accumulating data suggest that long non-coding RNA (lncRNA) H19 and p53are closely related to the prognosis of lung cancer. This study aims to analyze the association and interaction betweenH19 and mutant p53 R175H in lung adenocarcinoma (LAC). Methods: Mutant-type (Mt) p53 R175H was assessed by using RT-PCR in LAC cells and 100 cases of LAC tissue samples for association with H19 expression. Western blot, RNA-pull down, immunoprecipitation-Western blot and animal experiments were used to evaluate the interaction between H19 and mtp53. Results: Mtp53 R175H and H19 were over-expressed in LAC tissues and cells, while H19 over-expression extended the p53 half-life and enhanced transcriptional activity. Combined with anti-p53, ShH19 can significantly inhibit tumor growth in vivo. Conclusions: H19 over-expression may induce the elevated expression of mtp53 and interact with mtp53, leading to LAC progression. In addition, the high expression of mtp53 R175H is associated with poor overall survival inpatients. The simultaneous inhibition of H19 and mtp53 may provide a novel strategy for the effective control of LAC clinically.

## 1. Introduction

Coinciding with the progression of gene sequencing technology, it was shown that long non-coding RNAs (lncRNAs) are pervasively transcribed in the genome [1]. Generally, lncRNAs function to regulate gene expression, and aberrant lncRNA expression leads to tumor oncogenesis and progression [2]. For example, the imprinted, maternally expressed mouse H19 gene is located in the vicinity of several imprinted genes on the distal part of mouse chromosome7 [3] and localized at chromosome 11p15.5 in the human genome [4], and it has been shown to play a significant role in mammalian development and tumorigenesis [5,6]. H19 alterations occur in the early stages of cancer development and are thought to be involved in protein translational dysregulation and genomic instability, leading to cell proliferation imbalance and tumor metastasis [6,7]. Other studies, including our study, have disclosed that H19 is up-regulated in numerous human malignancies, such as intestinal [6], esophageal [8], bladder [9], breast [10], gastric [11] and lung cancer [7,12]. The functions of H19 can be dissected into two major types, i.e., the reservoir of miR-675 to suppress its targets [12] and the modulation of microRNAs and proteins through their binding capacity. P53 protein is a very important binding protein in tumors and the Tp53 gene is involved in gene regulation and cell apoptosis, and mutations are one of the frequent events in NSCLC [13,14,15,16]. Therefore, we searched for acorrelation between H19 and p53 and prognosis in 450 patients with lung cancer from The Cancer Genome Atlas (TCGA) database, and the results showed that there was a significant difference in overall survival between TP53 (+) H19 High and TP53 (−) H19 Low groups (*p* = 0.017, Figure 1A). Moreover, as can be seen from the TGCA database, the expression of H19 was significantly different between the TP53 (+) group and the TP53 (−) group (*p* = 0.003, Figure 1B).

However, these are only preliminary exploration results, and the actual driving mechanism is not clear. In our previous study, the mutation rate of p53 was 93%, and the most important mutation site was R175H [17]. Does H19 also play a regulatory role in p53 mutation? In this study, we explored the role and increased the understanding of H19 and mtp53 interactions in the promotion of LAC development and progression in vivo and in vitro assays.

## 2. Materials and Methods

### 2.1. Tissue Specimens

In the current study, samples were retrospectively collected from a total of 100 patients with LAC who underwent tumor resection at the Shanghai Ninth People’s Hospital, affiliated with Shanghai Jiaotong University (Shanghai, China) between 1 December 2012 and 12 December 2016. All of the patients were histologically diagnosed with LAC and classified according to the lung cancer TNM stages (the eighth edition) [18]. Tumor samples were taken from the surgery room during surgical resection of tumor lesions, snap-frozen in liquid nitrogen and stored at −80 °C. None of the included patients received any pre-surgery chemoradiotherapy. Patients who received preoperative chemotherapy or radiation therapy prior to surgery were omitted from this study. The research was approved by the Research Ethics Committee of Shanghai Jiaotong University (Shanghai, China) and Fudan University Shanghai Cancer Center, and written informed consent was obtained from all of the incorporated patients. The clinicopathological data of patients were collected from their medical records and are displayed in Table 1.

All of the patients were followed up regularly at 3–6-month intervals for 5 years and 6-month intervals thereafter. The last follow-up assessment was performed in December 2021 with a median follow-up duration of 65 months (ranging from 62 to 75 months). The overall survival was evaluated from diagnosis to the death of the patient.

### 2.2. Quantitative Reverse Transcriptase-Polymerase Chain Reaction (qRT-PCR)

Total cellular RNA was isolated from LAC cell line VMRC-LCD (p53 R175H mutation) and BEAS2B (normal lung cell) using a TRIzol reagent (Invitrogen, Carlsbad, CA, USA) and reverse transcribed into cDNA using a TAKARA kit (Invitrogen) according to the manufacturers’ protocols. The level of H19 expression in LAC cells was analyzed using qRT-PCR according to a previous study, while U6 mRNA was used as a loading control (5′-CAAATTCGTGAAGCGTTCCATAT-3′ and 5′-CAAATTCGTGAAGCGTATCTAT-3′) [10]. The level of p53 was amplified using qPCR with p53 primers (5′-GCCTGCAGTCCTGAGGTGTAGACGCCAAC-3′ and R5′-CGGAATTCTTGGTCTCCTCCACCGCTTCTT-3′). The ABI Step One Plus (Applied Biosystems, Foster City, CA, USA) was used to perform qRT-PCR amplification and the qRT-PCR data were quantified by using the 2^−ΔΔCt^ method. Each experiment was performed in triplicate and repeated at least once.

### 2.3. Cell Culture

Human LAC cell lines A549 (p53 wild type) and VMRC-LCD (p53 R175H mutation) were originally obtained from the American Type Culture Collection (Manassas, VA, USA) and maintained in Dulbecco’s modified Eagle’s medium (DMEM) supplemented with 10% fetal bovine serum (FBS, Invitrogen, Carlsbad, CA, USA), penicillin (100 U/mL, Sigma Chemicals, St. Louis, MO, USA) and streptomycin (100 µg/mL, Sigma Chemicals) in a humidified incubator with 5% CO_2_ at 37 °C.

### 2.4. H19 siRNA, Construction of H19 Expression Vector, and Cell Transfection

H19 cDNA was amplified using PCR with the TaKaRa LA Taq enzyme from A549 cells and the PCR products were then subcloned into pcDNA3.1 (Invitrogen, Carlsbad, CA, USA) between Kpn I and Xho I sites. The pcDNA3.1-19 cDNA was then DNA-sequence-confirmed and used to transfect LAC cells, while an empty pcDNA3.1 vector was used as a negative control. Moreover, H19 siRNA and negative control siRNA were purchased from Invitrogen and used to knockdown H19 expression in LAC cells.

For cell transfection, A549 cells were grown and transfected with pcDNA3.1 and pcDNA3.1-H19 or a vector-only control using Lipofectamine 2000 (Invitrogen) for 48 h and then selected in G418-containing growth medium at a stable cell population. The cells were then subjected to RNA or protein extraction or further functional experiments. For H19 siRNA transfection, tumor cells were grown and transfected with 5 μL of lipidosome H19 siRNA (5′-CCAACAUCAAAGACACCAUdTdT-3′) or a non-specific control siRNA (si-NC; purchased from TianGen) using Lipofectamine 2000 reagent for 15 min at room temperature. The cells were then assessed using qRT-PCR or other functional assays. The p53 expression was tested after being transfected with pcDNA3.1-H19 and a vector-only control or H19 siRNA and si-NC.

### 2.5. Cell Viability Assay

Cell viability was assayed by using a Cell Counting Kit-8 kit (CCK8; Beyotime, Shanghai, China). In brief, A549 and VMRC-LCD cells were plated into 96-well plates at a density of approximately 2 × 10^4^ cells per well and transiently transfected with pcDNA3.1-H19, siRNA-H19 and their controls for 48 h. Cells were then subjected to CCK-8 assay following the manufacturer’s instructions and the optical values of cells per well were measured using a SpectraMax^®^i3x microplate reader (Molecular Devices, Sunnyvale, CA, USA) at the absorbance of 450 nm. The experiments were performed in triplicate and repeated at least three times.

### 2.6. Colony Formation Assay

The H19-transfected LAC cells were seeded into six-well dishes containing pre-laid agarose at a density of 1 × 10^3^ and grown for two weeks. During this period, the growth medium was refreshed every three days. At the termination of each experiment, the cell colonies were stained with 0.5% crystal violet solution and counted under an inverted microscope (Olympus, Tokyo, Japan).

### 2.7. Western Blot Analysis of p53 Protein Half-Life

The gene-transfected LAC cells were grown and treated with 100 µg/mL of cycloheximide (Tiangen) for up to 6 h, and the total cellular protein content was lysed using the RIPA buffer and quantified for a Western blot analysis of the p53 level. Before cell collection, 100 μg/mL of cycloheximide was added to the 6-well plate for 0 h, 1 h and 6 h. Protein samples were collected at the same time 48 h after transfection. The protein samples were collected, and the content of target protein p53 in different dosing times was detected using Western blot. For Western blot assay, 50 µg protein samples were separated in sodium dodecyl sulfate-polyacrylamide gel electrophoresis (SDS-PAGE) gels and transferred onto polyvinylidene fluoride membranes (PVDF; Millipore, Billerica, MA, USA). The membranes were then blocked in 5% dried milk solution in PBS at room temperature for 2 h and incubated with a primary antibody at 4 °C overnight. The primary antibodies that were used were mouse monoclonal antibodies (Tiangen, Beijing, China) at a dilution of 1:100. The membranes were subsequently incubated with a horseradish peroxidase (HRP)-conjugated secondary antibody against mouse animal IgG (Tiangen) at room temperature for 1 h and then subjected to enhanced chemiluminescence (ECL) kit (Tiangen) incubation and exposed to X-ray films.

### 2.8. Luciferase Reporter Assay

The pGL3M-enhancer and p53-luc vectors were obtained from Tiangen (Beijing, China) and used to assess luciferase activity. The pGL3-enhancer vector contains an SV40-enhancer located downstream of luc+ and the poly A signal. It assisted in the verification of the functional promoter elements. This is why the presence of an enhancer will frequently result in the transcription of luc+ at higher levels. The p53-luc vector contained a p53 response element and a luciferase reporter gene at positions 32 nt~85 nt and 168 nt~1830 nt, respectively. For the luciferase reporter assay, VMRC-LCD cells were grown in 24-well plates and transfected with pcDNA3.1 and pcDNA3.1-H19, plus pGL3M-enhancer and p53-luc vectors using lipofectamine 2000 (Invitrogen) for 24 h according to the manufacturer’s instructions. Next, the total cellular protein was extracted using the RIPA buffer and quantified. The total protein was then subjected to the luciferase reporter assay using a luciferase dual assay kit from Promega (Madison, WI, USA) and the luciferase activity was measured using a Promega Glomax machine. The data are presented as the mean ± s.d. of triplicate data.

### 2.9. Biotin-RNA Pull-Down-Western Blot Assay

A pcDNA3.1-H19 plasmid was first linearized using XhoI enzyme, purified, and transcribed in vitro using a Qiagen kit (Qiagen, Berlin, Germany) tagged with biotin. The product of the transcribed RNA was purified using an H19A-clearTM kit (Ambion, Austin, TX, USA) following the manufacturer’s protocol. For the biotin–RNA pulldown assay, the gene-transfected LAC cells were lysed in radioimmunoprecipitation assay (RIRA) buffer containing 50 mM Tris, 1% Trition X-100 and 10% sodium dodecyl sulfate (SDS) on ice for 15 min followed by centrifugation and quantification. Cells were then subjected to Streptavidin bead pulldown (Aimeijie, Wuhan, China) for 1 h. Next, the protein samples were incubated with 1 mg of the biotin–RNA probe for 1 h at room temperature and then washed with phosphate-buffered saline (PBS) three times, boiled and subjected to a Western blot analysis of the p53 protein.

### 2.10. Immunoprecipitation-Western Blot

Gene-transfected LAC cells were washed in 2 mL of ice-cold PBS, lysed in 500 µL of the RIPA buffer on ice for 15 min and centrifuged at 10,000 rpm at 4 °C for 10 min, and the supernatant was quantified. For the RNA binding protein immunoprecipitation assay (RIP), the protein samples were incubated with 5 μg of the anti-p53 antibody for 2–16 h at 4 °C, during which time they were gently mixed. Next, the mixture was washed three times with PBS. Through centrifugation, the mixture was suspended on PBS. The suspension was incubated with 20 µL of protein A/G agarose (Thermo Scientific, Beijing, China) for 2 h at 4 °C, during which time it was gently mixed. The mixtures were then further washed with PBS three times through centrifugation. The resulting samples were resuspended in 50 μL of the SDS sample buffer, boiled and subjected to a Western blot analysis of the p53 level. The input protein samples were used as a negative control.

### 2.11. Xenograft Tumor Model

Six-week-old BALB/c nude mice were supplied by the Shanghai Jiaotong University Experiment Animal Center. The animal experiment program was in accordance with the Guidelines for the Care and Experimentation of Laboratory Animals from Shanghai Jiaotong University, and all the rules depicted in this research were approved by the Animal Care Committee of Shanghai Jiaotong University. ShH19 was acquired from Shanghai Tiangen Co., Ltd. (Shanghai, China). VMRC-LCD-shH19 xenografts were established (Shanghai Jiao Tong University Animal Laboratory Center) by inoculating 1 × 10^7^ cells (together with 50% Matrigel, BD Biosciences) into the abdominal mammary fat pad. Thirty nude mice suffering VMRC-LCD tumor xenografts were divided into three groups randomly: control group (1), shH19 group (2) and shH19 + anti-p53 group (3). In the third group, we applied the anti-p53 drug pifithrin(PFT) (2.2 mg/kg intraperitoneal injection) mixed with shH19. The anesthetic method of nude mice is 1% pentobarbital sodium I.P., and the dose for 20–25 g nude mice is generally 0.3 mL. The volumes of the tumors were measured every week. Nude mice were euthanized by pure carbon dioxide anesthesia.

### 2.12. Hematoxylin-Eosin (HE) Staining

The tumor tissues were paraffin-fixed and embedded in blocks of formalin-fixed tumor specimens. They were then subjected to routine HE staining and observed through optical microscopes. All the figures were obtained at a magnification of 400×.

### 2.13. Statistical Analysis

All statistical analyses used the SPSS 20.0 software (IBM, Armonk, NY, USA). The significance of the differences between the two groups was evaluated using the Student’s *t*-test or the χ^2^ test, as appropriate. The Kaplan–Meier curve and the log rank test were used to determine the overall survival stratified by the p53 expression. A two-sided *p*-value < 0.05 was considered to be statistically significant.

## 3. Results

### 3.1. Up-Regulated Mtp53 Expression in LAC Tissues and Cell Lines Associated with Poor Prognosis of LAC

Some preliminary results from TCGA data show that H19 and p53 have a certain correlation with survival rate and expression level (Figure 1A,B). Moreover, in our previous study, 100 NSCLC specimens were sequenced, in which 15 mutations occurred in codon 175 [17]. We performed PCR on these 15 cases, and we found, compared with normal lung tissues, mtp53 and H19 expression significantly increased, and the same result was obtained in the LAC cell (VMRC-LCD VS BSAS2B) experiment (Figure 1C,D). We found that mtp53 expression was elevated in both LAC tissues and cells with R175H mutation (Figure 1C,D). Meanwhile, A549 cells were subjected to PCR to evaluate the expression of wild-type p53 (Figure 1E). Additionally, in this cohort of patients with LAC, the total 5-year OS was 66%. At the last follow-up stage, 34 participants died. Among the patients who died, 20 were in stage IV (OS, 25.9%), 8 were in stage III (OS, 66.7%), 5were in stage II (OS, 78.2%), and 1 was in stage I NSCLC (OS, 96.2%) (Figure 1F,G). R175H mutations were significantly higher in men, smokers and advanced stages (Table 1). Notably, in our study, we found that mtp53 (R175H) expression was up-regulated and associated with poor prognosis in LAC, which was significantly different from other mutant sites (Figure 1F,G).

### 3.2. H19 Induction of LAC Cell Viability and Colony Formation of LAC Cells

Next, we transiently transfected H19 cDNA into A549 and VMRC-LCD cell line and performed a cell viability CCK-8 assay. We over-expressed H19 in A549 and VMRC-LCD cells (Figure 2A), and the CCK8 assay revealed that H19 over-expression distinctly accelerated the proliferation of A549 and VMRC-LCD cells (Figure 2B,C). The measured H19 transfection efficiency was 58% ± 5% (*n* = 10). Our data show that tumor cell viability and colony formation were significant in A549 and VMRC-LCD cell lines in which H19 was overexpressed (*p* = 0.001; Figure 2D).

### 3.3. Over-Expression or Inhibition of H19 on p53 Protein Half-Life in LAC Cells

The underlying cause of the higher level of p53 expression in LAC cells than in normal cells was analyzed through Western blotting analysis of the p53 protein (Figure 3A,B), after A549 and VMRC-LCD were transfected with pcDNA3.1-H19. As shown in Figure 3C,D, after the cells were treated with cycloheximide in different groups, the half-life of mtp53 was detected. We found that, compared toA549 cells, the half-life of p53 was prolonged in the over-expression of the H19 group in the VMRC-LCD cells, and if H19 was inhibited, the half-life of p53 was shortened. This may indicate that the over-expression of H19 prevents the degradation of mutant p53 (Figure 3C,D). There was hardly any difference in p53 levels between 1 h and 6 h time points for both the si-H19 and H19 conditions. This may imply that the interaction time between p53 and H19 predominantly occurs within one hour. A luciferase reporter assay was then performed using the H19 and p53-luc in VMRC-LCD cells. The results showed that luciferase activity was dramatically higher in the H19 group than in the pcDNA-3.1 group (*p* = 0.002; Figure 3E). Similarly, if H19 was inhibited, the luciferase activity was lower in the si-H19 group than in the si-NC group (*p* = 0.001; Figure 3E). In our study, the knockdown efficiency for H19 siRNA was 71.6%. H19 over-expression can also significantly enhance the transcriptional activity of mtp53.

### 3.4. H19 Interaction with Mtp53 in LAC Cells

To explore the interaction between H19 and mtp53, H19 was overexpressed in VMRC-LCD cells, and the level of mtp53 was assessed after that. The results revealed the mtp53 expression level was higher in H19-transfected cells than in the vector-only control group and was lower after H19 was knocked down (Figure 3F). For the further biotin–RNA pulldown test, a negative pair of radiator pcDNA3.1-GAPDH was constructed first. The pcDNA3.1-H19 and pcDNA3.1-GAPDH plasmids were linearized and purified, respectively. During in vitro transcription, the RNA was labeled with biotin–UTP, and the RNA-interacting proteins were obtained by virtue of the strong affinity and binding force between biotins and chain avidins. As shown in Figure 4A, in VMRC-LCD cells, H19 RNA can enrich the P53 protein. To exclude the possibility that the intracellular p53 protein complex interacts with H19, we performed in vitro experiments in which purified p53-GST was incubated with an H19 cRNA probe tagged with biotin, which still showed that H19 can also enrich free p53 protein (Figure 4A). However, we did not find that H19 was enriched for p53 protein in A549 cells (Figure 4B); that is, H19 was enriched only for mutant p53 (R175H) but not wild-type. We found that H19 was able to pulldown the p53 protein. The above results are the interaction between the H19 and p53 detected in vitro. Moreover, we performed an immunoprecipitation assay on VMRC-LCD cells, which showed that mtp53 could bind to H19 (Figure 4C). Negative results occurred in the A549 cell group (Figure 4D). After that, qRT-PCR analysis of the level of H19 in these samples revealed that the level of H19 was higher in VMRC-LCD cells than in the control (Figure 4E). However, no increase in H19 expression was observed in the A549 cell group (Figure 4F). The above results confirmed that H19 can indeed directly interact with p53.

### 3.5. ShH19 and Anti-p53 Inhibited Tumor Growth of Lung Cancer In Vivo

Thirty nude mice bearing VMRC-LCD tumor xenografts were generally randomly divided into three groups: control group, shH19 group and shH19 + anti-p53 group. Compared with the control group, the tumor growth inhibition was most obvious in the shH19 + anti-p53 group, followed by the shH19 group (Figure 5A). That is to say, the shH19 + anti-p53 combination had the most significant tumor inhibition effect. In addition, HE staining was performed after the mice were sacrificed. The tumor cells of the xenograft dramatically decreased in mice that received shH19+ anti-p53 in vivo compared to the control group (Figure 5B). Moreover, the PCR result showed the p53 expression was lower in the shH19 group than in the control group (Figure 5C), and the H19 expression was lower in the shH19 group and the shH19 + anti-group compared to the control group (Figure 5D).

## 4. Discussion

Non-coding RNAs are a class of RNA molecules that lack the ability to be translated into proteins [19]. According to the length (bp) of non-coding RNAs, they can generally be divided into several groups [20]: miRNAs, piwi-interacting RNAs (piRNAs), and lncRNAs. The expression of H19 in lncRNA is altered in various cancer types [21,22,23,24,25]. The correlation between H19 over-expression and lung cancer was reported in our previous studies [4]. Furthermore, wild-type p53 is the most important and well-studied tumor suppressor gene [26,27]. P53 is frequently mutated in lung cancer, and in our previous study, 93 of the 100 patients had p53 mutations, including 15 R175H [13,14,15,17]. In this study, we found that p53 was highly expressed in patients with R175H mutations and associated with poor LAC prognosis. Mtp53 R175H expression was associated with advanced clinicopathological features in LAC, such as an advanced NSCLC clinical stage, tobacco smoking and being male, which may be due to men smoking more. These data further confirm the findings of previous studies [7,28]. It has been confirmed that tobacco smoke can lead to the mutation of p53 [29], and one in seven cancer cases is caused by tobacco smoke [30]. The poor prognosis of the patients with a high expression of mtp53 was mainly due to the induction of the p53 mutation caused by the high expression of H19, and we clearly saw that wild-type p53 and other mutations were not for this. Therefore, we believe that the over-expression of H19 is an important cause of p53 mutations, resulting in some oncogene effects.

In this study, after the transfection of H19 into VMRC-LCD cells, cell proliferation and cell activity significantly increased, and cell colony formation was more intensive. The same results were obtained in both wild-type and mutant p53 lung cancer cell lines, which also indicated that the high expression of H19 can induce mutation inp53. In addition, we found that the expression of mtp53 was significantly increased after H19 over-expression and decreased after H19 inhibition. Therefore, it is clear that H19 has a close interaction with mutant p53.

Next, we explored the mechanism of action between H19 and mtp53. VMRC-LCD is the mtp53 R175H NSCLC cell line, which was our main research object. To date, there have been many studies investigating the role of p53 in the regulation of and interaction with other genes in lung cancer [13,14,15]. However, those were studies on H19 and wild-type p53, while our study focused on mtp53 R175H, which is the most common mutation site in lung cancer, revealing a poor prognosis for lung cancer and having a strong translational oncology value. In this study, we revealed an interesting and strong correlation between the status of the mtp53 gene and H19 RNA-induced expression. Up-regulations of H19 may combine and induce p53 mutations or participate in important mutation-induced regulation. Moreover, we found that the expression level of p53 was higher in H19-overexpressing A549 and VMRC-LCD cells. The high expression of H19 may induce the mutation of wild-type p53 to mutant p53. This work could lead to the development of novel therapeutic treatments for lung cancer.

The half-life of the wild-type p53 protein is awfully short, and some p53 mutations result in an increase in the half-life of p53 protein in cells; thus, detectable levels of p53 protein in cells or tissues usually represent mutant p53 (mt-p53) [31,32]. Indeed, our current study revealed that there are two possible causes of H19 over-expression-induced p53 expression: first, H19 induced p53 gene transcription and, second, an extension in the p53 protein half-life. The results showed that H19 both enhanced transcriptional activity and altered mtp53 protein degradation to extend the half-life, which is supported by our Western blotting data (Figure 3A–D). No increase in the p53 protein level was observed in H19 cDNA-transfected VMRC-LCD cells that were treated with cycloheximide to block protein translation. This indicates that the over-expressed of H19 interacts with mtp53 by prolonging its half-life rather than increasing p53 transcription, but this was negative on the A549 cells (wt-p53). This proves that H19 can interact with mutant p53 (R175H). The elucidation of this mechanism is beneficial to the treatment of LAC; that is, anti-p53 must be combined with anti-H19, which was confirmed in our subsequent animal experiments.

Previous studies have shown that lncRNAs mainly interact via proteins, such as the TATA-binding protein (TBP), to form an RNA–protein complex and, in turn, to change the structure and activity of the protein to control the biological functions in cells [33]. The TBP is thus a link between RNA and other proteins [34]. MtP53 protein is an important TBP. In our current study, our ex vivo data demonstrated that both p53 and H19 expressions were higher in LAC tissues and cells than in control cells and that mtp53 expression was associated with the H19 levels. In addition, we performed in vitro experiments in which H19 cRNA tagged with biotin was transcribed as a probe to pulldown H19-binding proteins and then performed a Western blot analysis of the p53 level and found that H19 did bind to the mtp53 protein in native or H19-transfected LAC cells (VMRC-LCD).However, this result was not seen in A549 cells. Moreover, we also immunoprecipitated the p53 protein using a p53 antibody and then performed qRT-PCR to identify H19 in the p53 protein complex. This further indicates that H19 binds to the mtp53 protein in LAC, and they could be combined together. However, we could not repeat the positive results in A549 (wt-p53) cells (Figure 4B,D,F). H19 does not have such strong links to wild-type p53 due to specific mutations in p53. In addition, H19 can not only bind to mutant p53 but also regulate the expression of mtP53 by enhancing its transcriptional activity and prolonging its half-life. By binding with p53 protein, the H19-mtp53 protein complex forms, which affects the function of the p53 protein, thus regulating downstream genes and the biological behavior of tumors. Similarly, in animal experiments, we found that shH19 expression can inhibit tumor growth, but anti-p53 + shH19 had a stronger inhibitory effect. Therefore, the simultaneous inhibition of H19 and P53 may have important value in the treatment of lung cancer.

## 5. Conclusions

In summary, our current study reveals the interaction between H19 and mtp53 in LAC carcinogenesis and development. It demonstrated that H19 over-expression may induce the elevated expression of mutant p53 and interact with mtp53, which extends the half-life and enhances transcriptional activity, leading to LAC progression. In addition, the high expression of mtp53 R175H is associated with poor overall survival inpatients. The simultaneous inhibition of H19 and p53 may provide a novel strategy for the effective control of LAC clinically.

## Figures and Tables

**Figure 1 cancers-14-04486-f001:**
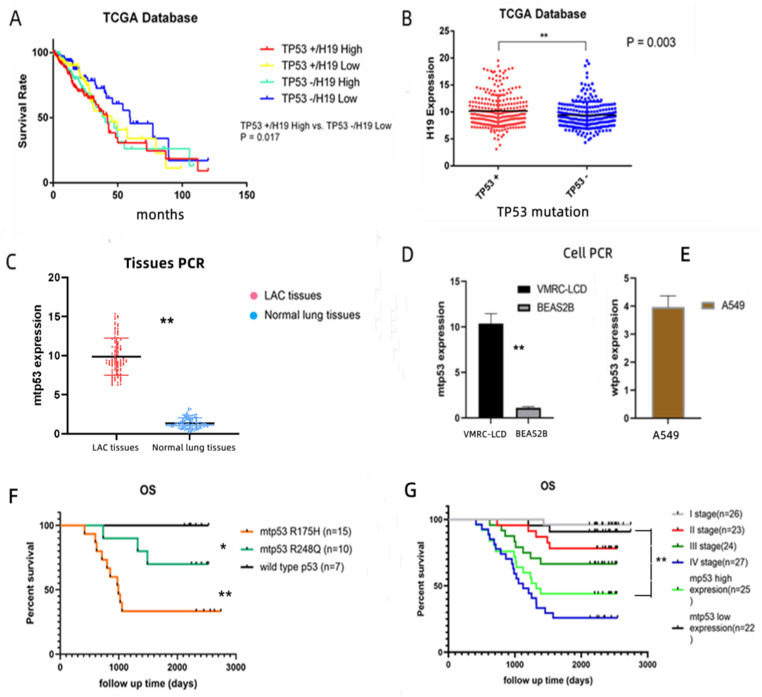
(**A**)The correlation between H19 and p53 and prognosis in 450 patients with lung cancer from The Cancer Genome Atlas (TCGA) database; the results showed that there was a marked difference in survival between the TP53(+)H19 High and the TP53(−)H19 Low groups (*p* = 0.017). (**B**) The expression of H19 was significantly different between the TP53 (+) group and the TP53 (−) group (*p* = 0.003). (**C**,**D**) Up-regulation of mtp53 (R175H) expression in LAC tissue specimens and cell lines (*p* = 0.01 and *p* = 0.002, respectively). (**E**) Wild-type p53 was slightly overexpressed in A549 lung adenocarcinoma cells. (**F**,**G**) Kaplan–Meier survival curves stratified by different mutant sites, mtp53 expressions and stages in patients with LAC. Mtp53 level was divided into low vs. high, which used the Ct value <5 vs. >5. Ct: cycle threshold. * (*p* < 0.05); ** (*p* < 0.01).

**Figure 2 cancers-14-04486-f002:**
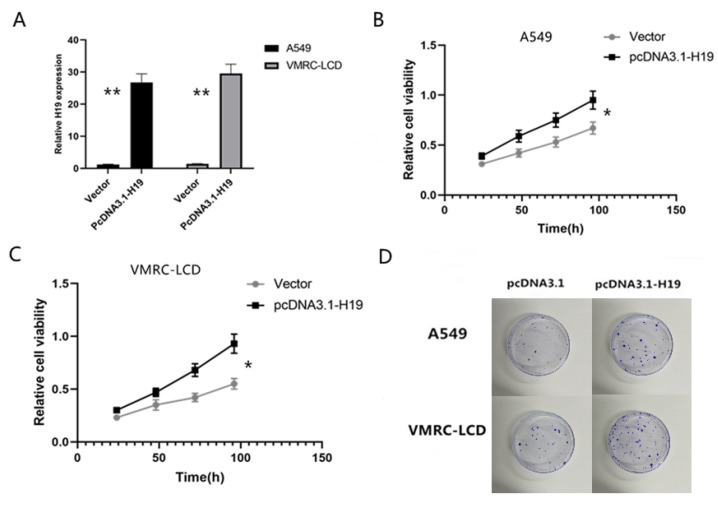
Effects of H19 over-expression on the regulation of LAC cell viability and colony formation. (**A**) H19 expression was significantly elevated after transfection with the vector in A549 and VMRC-LCD cells. (**B**,**C**) CCK8 assays showed that H19 over-expression promoted A549 and VMRC-LCD cell viability. (**D**) Colony formation assay results. The H19 cDNA-transfected A549 and VMRC-LCD cells were grown and subjected to the colony formation assay. The experiments were in triplicate and repeated at least three times. * (*p* < 0.05); ** (*p* < 0.01).

**Figure 3 cancers-14-04486-f003:**
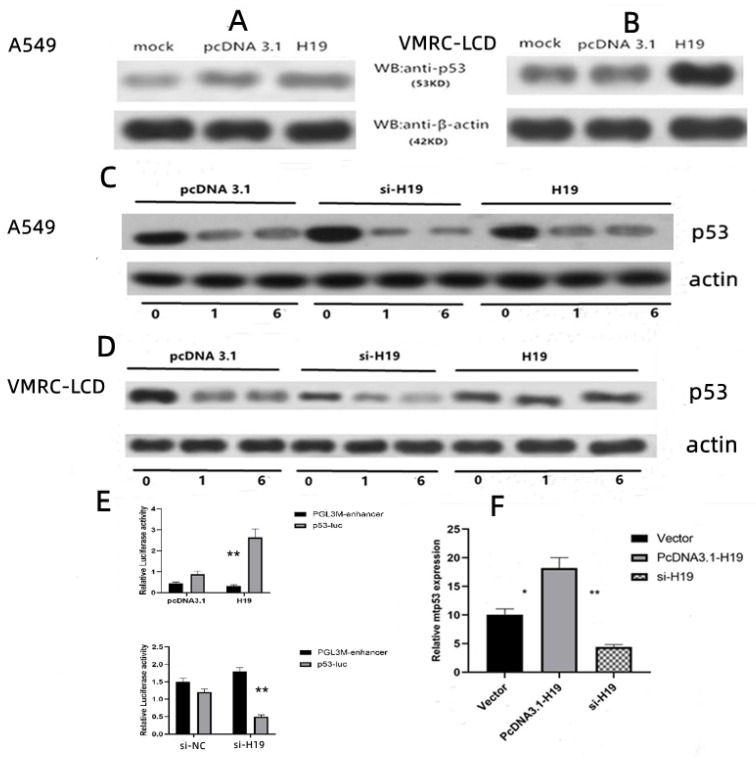
Effects of H19 over-expression on the regulation of mtp53 expression and activity. (**A**,**B**) Western blot results. NSCLC A549 and VMRC-LCD cells were grown and transiently transfected with pc-DNA3.1-H19 and subjected to a Western blot analysis of p53 protein. Concentration appears in the VMRC-LCD cells. (**C**,**D**) After transfection of si-H19 or pcDNA3.1-H19 in A549 and VMRC-LCD cells, 100 μg/mL of cycloheximide was added before cell collection for 0 h, 1 h and 6 h, and protein samples were collected. Compared toA549 cells, the half-life of p53 was prolonged in the over-expression of H19 in the VMRC-LCD cells. (**E**) Results of luciferase reporter assay. VMRC-LCD cells were grown and transiently co-transfected with H19 and p53-luc and subjected to a luciferase reporter assay. Luciferase activity was dramatically higher in the H19 group than in the pcDNA-3.1 group (*p* = 0.002). Similarly, if H19 was inhibited, luciferase activity was lower in the si-H19 group than in the si-NC group (*p* = 0.001) (**F**) After H19 was over-expressed the level of mtp53 was higher than the control group (*p* = 0.02). After H19 was knocked down with siRNA-H19, the expression of mtp53 was lower than in the control group (*p* = 0.024). * (*p* < 0.05); ** (*p* < 0.01). The uncropped blots are shown in Appendix A.

**Figure 4 cancers-14-04486-f004:**
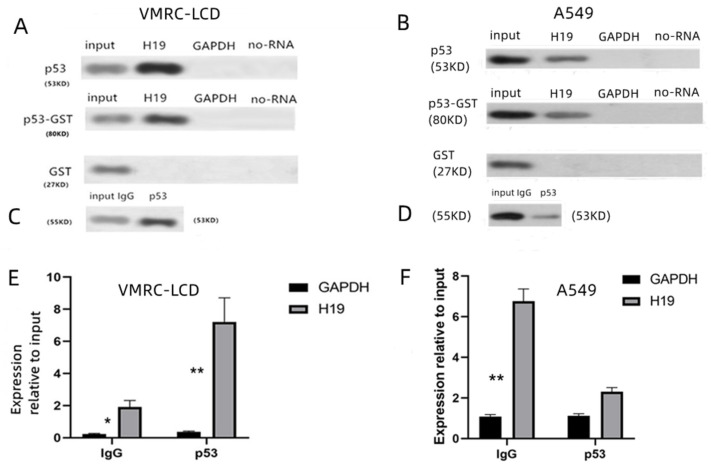
Interaction between H19 andmtp53 proteins. (**A**) In vitro pulldown assay to test if lncRNA H19 and mtp53 R175H interact with each other. VMRC-LCD cells were grown and transiently transfected with H19 for 48 h and then subjected to an RNA pulldown assay with an H19 cRNA probe and a Western blot analysis of the mtp53 protein. The H19 over-expression group showed concentration, suggesting that H19 binds to mtp53 (R175H). It was purified (GST) but still concentrated. (**B**) In vitro pulldown assay to test if lncRNA H19 and wild-type p53 interact with each other. A549 cells were grown and transiently transfected with H19 for 48 h and then subjected to an RNA pulldown assay with an H19 cRNA probe and a Western blot analysis of the p53 protein. Compared with the input, the concentration of the H19 over-expression group was decreased, indicating that H19 did not bind to wt-p53. After purification, there was still no concentration. (**C**,**D**) LncRNA H19 RNA immunoprecipitation was followed by Western blot to better show, H19 and mtp53 R175H interacted with each other. VMRC-LCD cells and A549 cells with p53 were grown and subjected to protein extraction and immunoprecipitation with an anti-p53 antibody and Western blot with a mouse IgG antibody to identify the protein–protein interaction. The results showed that mtp53 could enrich H19 and wild-type p53 does not bind to H19. (**E**,**F**) After the endogenous p53 protein and RNA complex were obtained and the RNA was extracted, qRT-PCR was used to detect the enrichment of H19 in the VMRC-LCD cells and A549 cells. The level of H19 was higher in the VMRC-LCD cells than in the control, and no increase in H19 expression was observed in the A549 cell group. * (*p* < 0.05); ** (*p* < 0.01). The uncropped blots are shown in Appendix A.

**Figure 5 cancers-14-04486-f005:**
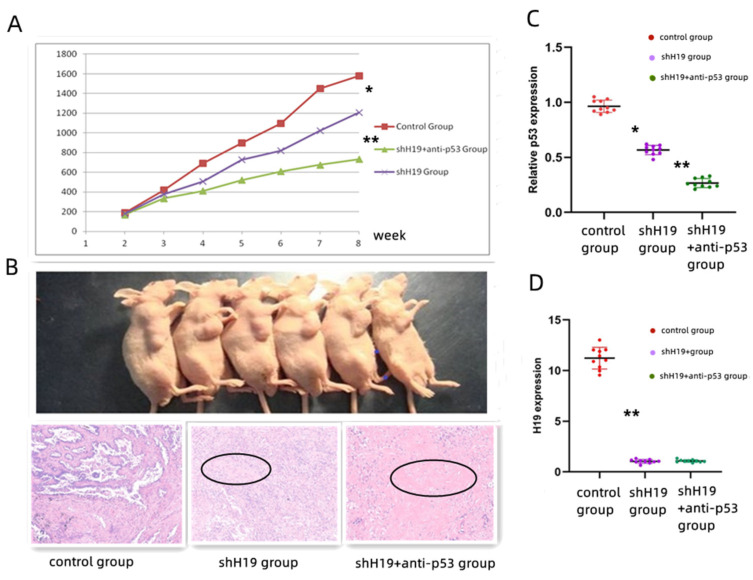
Animal experiments in vivo. (**A**) The tumor growth inhibition was most obvious in the shH19 + anti-p53 group, followed by the shH19 + anti-p53 group. (**B**)We chose two mice from each group as representatives, from left to right. The results of HE staining showed that, compared with the control group, the shH19 group began to show obvious tumor cell necrosis and fibrosis (black circles), which was more obvious in the shH19 + anti-p53 group. (**C**) The p53 expression was lower in the shH19 group than that in the control group (*p* = 0.02 and *p* = 0.001, respectively). (**D**) The H19 expression was lower in the shH19 group and shH19 + anti-group compared to the control group (*p* = 0.001). * (*p* < 0.05); ** (*p* < 0.01).

**Table 1 cancers-14-04486-t001:** Association of p53 expression with clinicopathological data from patients with LAC.

Clinicopathologic Data	N = 100	*p*
Age (year)		
<60	68	
≥80	32	
Gender		0.03
Male	58	
R175H	11	
Female	42	
R175H	4	
Tobacco smoke		0.02
Yes	49	
R175H	10	
No	51	
R175H	5	
P53 Status		
Wild type	7	
Mutant type	93	
Major mutation site		
R175H	15	
R248Q	10	
R249	7	
273	5	
282	4	
Others		
TNM stage		0.01
I	26	
R175H	0	
II	23	
R175H	2	
III	24	
R175H	5	
IV	27	
R175H	8	
Tumor differentiation		
Well/Moderate	37	
Poor	63	
R175H	15	

## Data Availability

The data used to support the findings of this study are available from corresponding authors upon request.

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
