# Peer review of "LncRNA H19 Promotes Lung Adenocarcinoma Progression via Binding to Mutant p53 R175H"

_cancers, 2022, doi:10.3390/cancers14184486_

Round 1
Reviewer 1 Report
In the manuscript titled “LncRNA H19 promotes lung adenocarcinoma progression via binding to mutant p53 R175H” investigates the relationship between lncRNA H19 and mtp53 (R175H) in lung adenocarcinoma. The authors observed that over-expression of H19 regulates p53 R175H to stimulate tumor cell progression. This is a well-performed study. The new findings can provide a novel strategy for the p53-R175H LAC. Thus, I recommend this manuscript for publication.
Author Response
Reviewer 1
In the manuscript titled “LncRNA H19 promotes lung adenocarcinoma progression via binding to mutant p53 R175H” investigates the relationship between lncRNA H19 and mtp53 (R175H) in lung adenocarcinoma. The authors observed that over-expression of H19 regulates p53 R175H to stimulate tumor cell progression. This is a well-performed study. The new findings can provide a novel strategy for the p53-R175H LAC. Thus, I recommend this manuscript for publication.
Thank you very much.
Reviewer 2 Report
Zhou and Xia have shown that lncRNA H19 binds to mutant p53 and promotes lung adenocarcinoma progression. The overall quality of data looks fair and supports the claim, but the paper is poorly written, and the data is not appropriately explained; therefore, it is hard to follow and understand. In addition, I have a few more comments which need to address to make the manuscript suitable for publication.
1. The first paragraph in the introduction section is redundant, and there is no need to write this in the paper. Please remove it.
2. Please provide significance tests and mark them on all the graphs. There are many instances where the significance test has not been shown. For example, Fig 1E-F, 3D-E, 4B, and 5A.
3. Fig 1A-B is not discussed in the result section.
4. Fig 1C, since this data comes from 100 NSCLC patients, please show it in a dot-plot graph.
5. Line 268 “The measured H19 transfection efficiency was…..” data not shown, please provide data.
6. Fig 2B-C, A549 cell line also shows increased cell viability upon H19 over-expression comparable to VMRC-LCD, although H19 binds to a lesser extent to wt p53 (based on the data shown in figure 3A). Please explain and provide a reason.
7. Fig 2D, this data also showed higher colony formation in A549 i.e., wt p53 compared to VMRC-LCD mtp53. Please explain and also provide quantification for Fig 2D and the significance test.
8. Fig 3C, why the expression level of p53 is low in H19 transfected compared to vector only at 0 hr? This data looks opposite to what is shown in figure 3B.
9. Fig 5A, what is on X-axis?
10. Fig 5B, please mark these mice for their groups. What do the authors want to show by H&E of tumors? It is not clear in terms of the objectivity of this experiment.
11. Fig 5C, please provide a dot-plot graph type as this data comes from 10 mice from each group; it would be easier to visualize. Please also provide the H19 expression.
Author Response
Reviewer 2
Zhou and Xia have shown that lncRNA H19 binds to mutant p53 and promotes lung adenocarcinoma progression. The overall quality of data looks fair and supports the claim, but the paper is poorly written, and the data is not appropriately explained; therefore, it is hard to follow and understand. In addition, I have a few more comments which need to address to make the manuscript suitable for publication.
1. The first paragraph in the introduction section is redundant, and there is no need to write this in the paper. Please remove it.
Thank you for your suggestions. I had removed the first paragraph.
2. Please provide significance tests and mark them on all the graphs. There are many instances where the significance test has not been shown. For example, Fig 1E-F, 3D-E, 4B, and 5A.
Ok. I had added the significance tests and mark them on all the graphs in Fig 1E-F, 3D-E, 4B, and 5A.
3. Fig 1A-B is not discussed in the result section.
Thank you for your advice. I added the discussion of Fig 1A-B in the result paragraph 3.1 in red.
4. Fig 1C, since this data comes from 100 NSCLC patients, please show it in a dot-plot graph.
I changed Fig 1C into in a dot-plot graph.
5. Line 268 “The measured H19 transfection efficiency was…..” data not shown, please provide data.
Thank you. Flow cytometry was used to detect and calculate the transfection efficiency, which is 58% ± 5%.
6. Fig 2B-C, A549 cell line also shows increased cell viability upon H19 over-expression comparable to VMRC-LCD, although H19 binds to a lesser extent to wt p53 (based on the data shown in figure 3A). Please explain and provide a reason.
Thank you for your very good questions. We speculated that H19 over-expression induce p53 mutation or increase the binding ability of H19 and p53. This still needs to be studied and confirmed by our further experiments.
7. Fig 2D, this data also showed higher colony formation in A549 i.e., wt p53 compared to VMRC-LCD mtp53. Please explain and also provide quantification for Fig 2D and the significance test.
Thank you. We speculated that H19 over-expression induce p53 mutation or increase the binding ability of H19 and p53. This still needs to be studied and confirmed by our further experiments.
8. Fig 3C, why the expression level of p53 is low in H19 transfected compared to vector only at 0 hr? This data looks opposite to what is shown in figure 3B.
Thank you for your question. Indeed, as you said, at 0 hours, there is no significant difference between the two. However, with the extension of time, the vector group began to decay, so the expression decreased. Figure 3B shows at least 6 hours, so it is not inconsistent with Figure 3C.
9. Fig 5A, what is on X-axis?
I am sorry and the X-axis is week.
10. Fig 5B, please mark these mice for their groups. What do the authors want to show by H&E of tumors? It is not clear in terms of the objectivity of this experiment.
Thank you for your advice. We chose two mice from each group as representatives, from left to right. The purpose of using HE is to compare the results of different treatments between different groups.
11. Fig 5C, please provide a dot-plot graph type as this data comes from 10 mice from each group; it would be easier to visualize. Please also provide the H19 expression.
Thank you for your suggestion. I revised the Fig 5C.
Reviewer 3 Report
The authors of the manuscript "LncRNA H19 promote lung adenocarcinoma progression via binding to mutant p53 R175H" have attempted to make a connection between lncRNA H19 with the mutant form of p53 R175H. However, the manuscript is not well written and organized. Many conclusions in the manuscript lack experimental evidence or are contradicted by authors own statements in the manuscript. Also the evidence to show specific effects of H19 on mutant p53 is very weak, since in most experiments H19 has the same effects in both p53 wt and mutant p53 cell lines. Therefore, I would suggest the authors to revise the manuscript well and attempt a resubmission.
Here are some of my major concerns:
1. Introduction line 36-44, do not belong in the manuscript and this start itself shows how poorly edited and organized the manuscript is. Abstract is also rather a summary and not suitable as an abstract.
2. All the figures quality is poor and size is small, with the legends hardly readable.
3. In figure 1c, it will be interesting to know the levels of H19 lncRNA in these normal vs. LAC tissue samples. In figure 2d, please also add A549 cell line p53 mRNA data, which is also used in the subsequent experiments in the manuscript as a cell line with WT p53?
4. In figure 2a, how does the p53 mRNA levels change upon H19 overexpression in both the A549 (wt p53) and VMRC-LCD (R175H p53)?
5. In figure 2b,c do the author mean that the cells grow faster upon H19 overexpression? Since cell viability is not a good indicator for cell proliferation/ growth as labelled on the y-axis? Also plot both the graphs (A549 and VMRC-LCD) as a single plot, since then readers can see whether the effects of H19 overexpression on cell proliferation are specific to mutant p53. Looking at the data as is it looks like H19 overexpression increases proliferation irrespective of whether the cell lines have a p53 mutation or not, which reduces the novelty of the manuscript and challenges the claims of authors that H19 specifically affects only mutant p53 biology.
6. Knockdown efficiency for H19 siRNA is missing in the manuscript.
7. In figure 3c, it is essential to have the same experiment and western blot also done in the A549 cells with wt p53, to know whether H19 regulates the protein half-life of mutant p53 specifically. The conclusions made in the manuscript in the current form are therefore not well supported due to lack of this essential control experiments in A549 cells with wt p53, in this figure and the remainder here on further.
8. In figure 4, the authors have attempted to identify potential interaction between mutant p53 and H19, however all the assays are done in A549 cells which have wt p53 protein. Is it a mistake in the writing, as in the current state, it looks that H19 binds wt p53 and the effects shown by authors could be generic effects of H19-wt p53 interaction and not r175h mutation specific?
9. In figure 5a, the x-axis label is missing and a important control for only anti-p53 treated mice is missing. How to authors rule out the possibility that the very strong effects seen in the shH19+ anti-p53 are not only from anti-p53?
10. The table 1, should be placed very early in the manuscript along with the patient data and not in the end.
I therefore suggest the authors to rework their manuscript and address these critical limitations before resubmission.
Author Response
Reviewer 3
The authors of the manuscript "LncRNA H19 promote lung adenocarcinoma progression via binding to mutant p53 R175H" have attempted to make a connection between lncRNA H19 with the mutant form of p53 R175H. However, the manuscript is not well written and organized. Many conclusions in the manuscript lack experimental evidence or are contradicted by authors own statements in the manuscript. Also the evidence to show specific effects of H19 on mutant p53 is very weak, since in most experiments H19 has the same effects in both p53 wt and mutant p53 cell lines. Therefore, I would suggest the authors to revise the manuscript well and attempt a resubmission.
Here are some of my major concerns:
1. Introduction line 36-44, do not belong in the manuscript and this start itself shows how poorly edited and organized the manuscript is. Abstract is also rather a summary and not suitable as an abstract.
Thank you for your suggestion. I deleted the first paragraph.
2. All the figures quality is poor and size is small, with the legends hardly readable.
I am sorry for that and revised the figures and legends.
3. In figure 1c, it will be interesting to know the levels of H19 lncRNA in these normal vs. LAC tissue samples. In figure 2d, please also add A549 cell line p53 mRNA data, which is also used in the subsequent experiments in the manuscript as a cell line with WT p53?
Thank you for your good questions. The levels of H19 lncRNA in these normal vs. LAC tissue sample were performed in our previous research and the expression of H19 was higher in LAC tissue samples. We performed the PCR of A549 and I added A549 cell line p53 mRNA in figure 1d.
4. In figure 2a, how does the p53 mRNA levels change upon H19 overexpression in both the A549 (wt p53) and VMRC-LCD (R175H p53)?
After H19 over-expression, p53 mRNA levels increased.
5. In figure 2b,c do the author mean that the cells grow faster upon H19 overexpression? Since cell viability is not a good indicator for cell proliferation/ growth as labelled on the y-axis? Also plot both the graphs (A549 and VMRC-LCD) as a single plot, since then readers can see whether the effects of H19 overexpression on cell proliferation are specific to mutant p53. Looking at the data as is it looks like H19 overexpression increases proliferation irrespective of whether the cell lines have a p53 mutation or not, which reduces the novelty of the manuscript and challenges the claims of authors that H19 specifically affects only mutant p53 biology.
Thank you for your analysis. In our view, H19 overexpression can lead to cell proliferation in the mutant p53 cell line, as well as in the A549 cell line, because H19 overexpression can induce wild-type p53 mutations.
6. Knockdown efficiency for H19 siRNA is missing in the manuscript.
I am sorry for the missing and added the knockdown efficiency (71.6%) in Page 6 in red.
7. In figure 3c, it is essential to have the same experiment and western blot also done in the A549 cells with wt p53, to know whether H19 regulates the protein half-life of mutant p53 specifically. The conclusions made in the manuscript in the current form are therefore not well supported due to lack of this essential control experiments in A549 cells with wt p53, in this figure and the remainder here on further.
Thank you for your suggestions. We preliminarily saw the difference between the two after H19 transfection in Figure 3A and 3B, so we selected the mutant p53 cell line to make Figure 3C. However, as you say, it is essential to have the same experiment and western blot also done in the A549 cells with wt p53. We are making up the experiment.
8. In figure 4, the authors have attempted to identify potential interaction between mutant p53 and H19, however all the assays are done in A549 cells which have wt p53 protein. Is it a mistake in the writing, as in the current state, it looks that H19 binds wt p53 and the effects shown by authors could be generic effects of H19-wt p53 interaction and not r175h mutation specific?
I am sorry for this mistake in legends. All the assays are done in VMRC-LCD cells.
9. In figure 5a, the x-axis label is missing and an important control for only anti-p53 treated mice is missing. How to authors rule out the possibility that the very strong effects seen in the shH19+ anti-p53 are not only from anti-p53?
Thank you for your advice. I added the x-axis. I think you are right about increasing the anti-p53 group. We had considered adding this group, but as we have seen in other literature, the inhibitory effect of p53 alone on tumor is not obvious. Thank you very much for your advice. We will make up the anti-p53 experiment later.
10. The table 1, should be placed very early in the manuscript along with the patient data and not in the end.
Thank you. Table 1 was first placed in paragraph 2.1.
I therefore suggest the authors to rework their manuscript and address these critical limitations before resubmission.
Reviewer 4 Report
LncRNA H19 promote lung adenocarcinoma progression via binding to mutant p53 R175H
Yaodong Zhou and Qing Xia
Summary:
Zhou and Xia present a story alluding to an interaction between lncRNA H19 and the mutant p53 (p53R175H).
Maybe this is true, and there is indeed an interaction that elicits an effect on lung cancer progression; but the authors have not brought that forth.
Major questions/concerns:
-
Fig. 1: How were the 450 patients selected? Out of how many lung adenocarcinoma patients on TCGA? What are the criteria for this selection?
-
How do the H19 and p53 levels compare across other cancer subtypes?
-
What are the levels of mutant p53 and lncRNA H19 from cell lines that have similar genotypes as the patients?
-
Fig. 2: 2A is just showing that the transient transfection works. 2B and 2C: Cell proliferation (as described in the figure legend) is not the same as cell viability. Please present cell numbers at times 0, 12, 24, 48, 72, and 96 hours for the two cell lines.
-
Fig. 2D is not readable at all!
-
Fig. 3: 3A and 3B: Quantifying the western blots by densitometry (Image J or similar) would make it more easily legible.
-
Fig. 3C: There is hardly any difference in p53 levels between 1-hour and 6-hour time points for both si-H19 and H19 conditions. This should be highlighted.
-
Fig. 3D and 3E: Were any statistical tests performed on these data to compare the effects to controls?
-
Fig. 4: The figure legend doesn’t explain the experiments at all. Fig. 4A is ambiguous at best, confusing and inconclusive. What was pulled down? What are the controls? Why is the Y-axis labeled “signal relative to input” in Fig. 4B? What is the signal? What is the input?
-
This should be replaced by a) In vitro pulldown assay to test if lncRNA H19 and p53 R175H interact with each other. b) lncRNA H19 RNA immunoprecipitation followed by western blot could better show - and conclusively - if H19 and p53 R175H interact with each other.
-
Fig. 5: What is “anti-p53 group”? It is not really defined.
Author Response
Reviewer 4
Summary:
Zhou and Xia present a story alluding to an interaction between lncRNA H19 and the mutant p53 (p53R175H).
Maybe this is true, and there is indeed an interaction that elicits an effect on lung cancer progression; but the authors have not brought that forth.
Major questions/concerns:
1. Fig.1: How were the 450 patients selected? Out of how many lung adenocarcinoma patients on TCGA? What are the criteria for this selection?
Thank you for your question. Our screening in TCGA database was mainly based on patients with different H19 and p53 expression in lung adenocarcinoma.
2. How do the H19 and p53 levels compare across other cancer subtypes?
In other types, the expression of H19 was increased but not high. The mutation rate of p53 in squamous cell carcinoma is high, but the main site is not R175H.
3. What are the levels of mutant p53 and lncRNA H19 from cell lines that have similar genotypes as the patients?
Figure1D showed levels of mutant p53 from cell lines VMRC-LCD and our previous research also showed that the expression of H19 lung adenocarcinoma cell lines such as A549 and H1299 was significantly increased.
4. Fig 2: 2A is just showing that the transient transfection works. 2B and 2C: Cell proliferation (as described in the figure legend) is not the same as cell viability. Please present cell numbers at times 0, 12, 24, 48, 72, and 96 hours for the two cell lines.
I am sorry for mistake in the figure2 legend. I revised them in red. (B and C) CCK8 assays showed that H19 overexpression suppressed A549 and VMRC-LCD cell viability
5. Fig. 2D is not readable at all!
I am sorry for that and this figure is not clear.
6. Fig. 3: 3A and 3B: Quantifying the western blots by densitometry (Image J or
similar) would make it more easily legible.
Thank you for your nice suggestions. I will quantify the western blots by densitometry (Image J).
7. Fig. 3C: There is hardly any difference in p53 levels between 1-hour and 6-hour time points for both si-H19 and H19 conditions. This should be highlighted.
Thank you for your advice. It may imply that the interaction time between p53 and H19 predominantly occurs within one hour. I added them in paragraph 3.3 in red.
8. Fig. 3D and 3E: Were any statistical tests performed on these data to compare the effects to controls?
Yes. As you said, we added the statistical tests and marked it with an asterisk.
9. Fig.4: The figure legend doesn’t explain the experiments at all. Fig. 4A is ambiguous at best, confusing and inconclusive. What was pulled down? What are the controls? Why is the Y-axis labeled “signal relative to input” in Fig. 4B? What is the signal? What is the input?
Thank you for your questions. P53 protein was pulled down. The controls are input IgG. “Signal relative to input” means an expression relative to input, and I changed into “expression relative to input”. The input refers to the p53 protein pulled down.
10. This should be replaced by a) In vitro pulldown assay to test if lncRNA H19 and p53 R175H interact with each other. b) lncRNA H19 RNA immunoprecipitation followed by western blot could better show and conclusively - if H19 and p53 R175H interact with each other.
Thank you for your excellent suggestions. According to your suggestion, I modified the legends of Figure 4.
11. Fig. 5: What is “anti-p53 group”? It is not really defined.
I am sorry for that. In paragraph 2.11, I denied “anti-p53 group” as “we applied the anti-p53 drug Pifithrin-(PFT) (2.2mg/kg intraperitoneal injection)”.
Round 2
Reviewer 2 Report
I have only two comments that need to be addressed before publication.
Regarding previous comment 6 and the authors response:
Since the main conclusion of this paper is that lncRNA H19 promotes lung adenocarcinoma progression by inducing elevated expression of mtp53 and binding to it, speculation would not be enough to say that lncRNA H19 causes p53 mutation. This needs a piece of evidence to support this statement. Hence, the data in Fig 2B-C are not conclusive in supporting that interaction of H19 with mutated p53 is responsible for increased cell viability.
Fig 5B. Please mark the tumor cell necrosis and fibrosis in the H&E images.
Reviewer 3 Report
In the revised version of the manuscript, the authors still have not sufficiently clarified some of the critical points from my first review. Below are some of the critical points which in my opinion the authors need to work on for this manuscript to be accepted.
1. In figure 2b and 2c, the authors say that the H19 overexpression leads to suppression of cell viability. However, this is clearly opposite to what the graphs show, wherein pcDNA3.1-H19 transfected cells clearly have higher relative cell viability than the vector control. This point was also mentioned in the first review and is still unclear and the conclusion is completely opposite to what the data shows.
2. The authors response:
"Thank you for your analysis. In our view, H19 overexpression can lead to cell proliferation in the mutant p53 cell line, as well as in the A549 cell line, because H19 overexpression can induce wild-type p53 mutations."
The above statement is a complete speculation and there is no supporting proof for this statement in the manuscript. Therefore, it remains still unclear why H19 overexpression has similar effects on wtp53 and mtp53? This distinction is central to the theme of the manuscript and needs clarification.
3. The authors response:
"Thank you for your suggestions. We preliminarily saw the difference between the two after H19 transfection in Figure 3A and 3B, so we selected the mutant p53 cell line to make Figure 3C. However, as you say, it is essential to have the same experiment and western blot also done in the A549 cells with wt p53. We are making up the experiment"
This experiment is again very important to show that there is a direct link only between mtp53 and H19 like the authors have concluded in the manuscript. This experiment needs to be added to support authors claims and for publication. Also this is important experiment, as it seems in figure 3a, that in A549 cells, H19 overexpression is also stabilizing the wtp53.
4. In figure4, it is important to perform the assays done with mtp53, in A549 cell line to show that H19 does not interact with the wtp53. This is again essential for the very strong conclusions made by the authors for the mtp53-H19 interaction.
Due to these above points, I still do not see the manuscript fit for publication as the conclusions are not very well supported yet. I recommend the authors to perform the above experiments to improve the impact of their publication.
Reviewer 4 Report
I do not see any substantial improvement in the manuscript from the previous submission - hence I will stick with my earlier opinion.
Author Response
I do not see any substantial improvement in the manuscript from the previous submission - hence I will stick with my earlier opinion.
I'm sorry to disappoint you. I added experiments and performed the assays done with mtp53 in A549 cell line. I revised figures 3 and 4. Many revisions and additions have been made to the results and legends sections of the manuscript in red. Thank you very much for your hard work in reviewing the manuscript. I gained a lot.
Round 3
Reviewer 3 Report
The authors of the manuscript "LncRNA H19 promote lung adenocarcinoma progression via binding to mutant p53 R175H" in this second revision have addressed most of the points raised in my last two revisions. The authors have provided the evidence for H19 overexpression leading to p53 mutations, though the experiments are not yet performed in replicates and not included in the manuscript with proper statistics. In my opinion, this is sufficient for the moment since the authors want to investigate this effect in subsequent publications.
However, this leads to a contradiction in their subsequent findings regarding the H19 overexpression in Figure 3c, where in A549 cells the overexpression of H19 does not lead to increased protein half-life for the potentially mutated p53 in these A549 cells? Would the authors kindly explain this potential discrepancy, since it also forms a major point in their discussion and mechanism.
Lastly, the authors need to proof-read the manuscript and the figure labels carefully, to indicate the mutant p53 (mtp53), wherever applicable. With these minor clarifications and proper proof-reading, I see the manuscript fit for publication.
Author Response
The authors of the manuscript "LncRNA H19 promote lung adenocarcinoma progression via binding to mutant p53 R175H" in this second revision have addressed most of the points raised in my last two revisions. The authors have provided the evidence for H19 overexpression leading to p53 mutations, though the experiments are not yet performed in replicates and not included in the manuscript with proper statistics. In my opinion, this is sufficient for the moment since the authors want to investigate this effect in subsequent publications.
However, this leads to a contradiction in their subsequent findings regarding the H19 overexpression in Figure 3c, where in A549 cells the overexpression of H19 does not lead to increased protein half-life for the potentially mutated p53 in these A549 cells? Would the authors kindly explain this potential discrepancy, since it also forms a major point in their discussion and mechanism.
Thank you for your kindness and help. As you said, we have also found this result. We also see a bit of lengthening protein half-life in Figure 3C, but it is significantly weaker than Figure 3D. This reasonably explains that although H19 overexpression can induce wild-type mutation and then interact with each other, the degree of binding is not as high as that of direct binding to mutant p53.
Lastly, the authors need to proof-read the manuscript and the figure labels carefully, to indicate the mutant p53 (mtp53), wherever applicable. With these minor clarifications and proper proof-reading, I see the manuscript fit for publication.
Thank you very much again. I will go over the full manuscript and the figures again.
Reviewer 4 Report
After these extensive rounds of revision, the manuscript warrants publication.
Author Response
After these extensive rounds of revision, the manuscript warrants publication.
Thank you very much for your help.